# Soluble Dietary Fiber from *Citrus unshiu* Peel Promotes Antioxidant Activity in Oxidative Stress Mice and Regulates Intestinal Microecology

**DOI:** 10.3390/foods13101539

**Published:** 2024-05-15

**Authors:** Mengqi Fu, Xin Gao, Zuorui Xie, Chenlan Xia, Qing Gu, Ping Li

**Affiliations:** Key Laboratory for Food Microbial Technology of Zhejiang Province, College of Food Science and Biotechnology, Zhejiang Gongshang University, Hangzhou 310018, China; fumengqi771a@163.com (M.F.); 18989459736@163.com (X.G.); xiezuorui2020@outlook.com (Z.X.); xiachenlan2004@outlook.com (C.X.); guqing2002@hotmail.com (Q.G.)

**Keywords:** oxidative, *Citrus unshiu* peel, dietary fiber, intestinal flora, mice

## Abstract

Aging is characterized by the progressive degeneration of bodily tissues and decline in physiological functions, a process that may be exacerbated by imbalances in intestinal flora. Soluble dietary fiber (PSDF) from *Citrus unshiu* peel has demonstrated strong free radical scavenging ability to regulate intestinal flora in vitro. However, further evidence is required to ascertain the effectiveness of PSDF in vivo. In our study, 8-week-old mice were artificially aged through subcutaneous injections of a 200 mg/kg/d D-galactose solution for 42 days, followed by a 28-day dietary intervention with varying doses of PSDF, insoluble dietary fiber (PIDF), and vitamin C. After the intervention, we observed a significant mitigation of D-galactose-induced oxidative stress, as evident by weight normalization and reduced oxidative damage. 16S rRNA gene sequencing revealed that PSDF significantly altered the composition of intestinal flora, increasing Firmicutes and reducing Bacteroidota percentages, while also enriching colonic short-chain fatty acids (SCFAs). Spearman correlation analysis further identified a positive correlation between Firmicutes and isovaleric acid, and negative correlations between *Muribaculaceae* and acetic acid, and between *Lachnospiraceae_NK4A136_group* and caproic acid. These findings support the potential of Citrus PSDF to alleviate oxidative stress.

## 1. Introduction

The scientific name of the Wenzhou mandarin is *Citrus unshiu*, which belongs to the genus *Citrus* within the Rutaceae family. It is a type of citrus fruit known for its thick rind. *Citrus unshiu* is the main citrus production variety in China, and the annual output in Zhejiang Province has reached hundreds of tons [1]. Remigante et al. found that bergamot extract can effectively prevent the production of reactive oxygen species induced by D-galactose, and alleviate the oxidative stress damage of biomacromolecules including membrane lipids and proteins [2]. Most of the peels are discarded as factory waste, which cannot be effectively used and even cause environmental pollution. Discarded citrus peels are rich in numerous active components. Given their easy availability and low cost, these waste materials can be repurposed as raw materials for reprocessing to enhance the overall added value. This approach not only promotes the efficient use of resources but also contributes to environmental sustainability by reducing waste [3]. “Chenpi”, a traditional Chinese herbal medicine, is made from the peels of citrus fruits (Rutaceae family) through a special sun-drying process. In Chinese medicine, Chenpi is known for its efficacy in resolving phlegm and strengthening the spleen, dispelling dampness, as well as treating asthma. This showcases the traditional utilization of citrus peels, aligning with the concept of repurposing waste materials for added value, but also highlights the medicinal properties attributed to these peels in historical practices [4].

Citrus peels contain a variety of functional active substances, among which dietary fiber accounts for more than 50% [5]. Dietary fiber (DF), a complex polysaccharide composed of multiple monomers, is primarily located in plant cell walls or derived from intracellular sources. Studies have demonstrated that sufficient intake of dietary fiber can reduce blood sugar levels, lipid concentrations, and cancer risk [6]. It can be categorized into soluble dietary fiber (SDF) and insoluble dietary fiber (IDF) based on its water solubility [7]. SDF has been found to exhibit a superior antioxidant capacity compared to IDF, likely due to increased oligosaccharide content and reduced molecular weight [8]. This antioxidant capability is assessed in vitro through free radical scavenging, metal ion reduction, and lipid peroxidation inhibition [9]. In vivo, the induction of oxidative stress and accelerated aging by excessive D-galactose injection, which elevates galactitol and reactive oxygen species (ROS) levels, serves as a model to evaluate substances’ antioxidant effectiveness [10]. The presence of high ROS levels boosts oxidase activity and malondialdehyde (MDA) content, while antioxidant enzymes like SOD and GSH-Px act as primary defenses against ROS, making their activity a key indicator of the body’s antioxidant capacity [11]. Despite the known benefits, it remains uncertain whether dietary fibers found in citrus fruits possess antioxidant activities within the body and how they affect oxidative stress. This area of research is crucial for understanding the full spectrum of health benefits that citrus dietary fibers may offer, particularly in the context of preventing or mitigating diseases associated with oxidative stress and inflammation.

Studies have shown that dietary fiber is closely related to the regulation of intestinal flora. Dietary fiber is low in bioavailability and cannot be broken down by humans’ own enzymes. Therefore, it ranks last among the several nutrients that humans need. The large microbiome in the human intestinal tract can use these substances for fermentation. The colon provides a favorable living environment for intestinal bacteria, containing more than 70% of the microbes in the human body [12]. When dietary fiber is decomposed by colon microbes, on the one hand, it can produce new small molecular carbohydrate; on the other hand, it can affect the composition of intestinal flora and metabolic enzyme activity, so as to regulate the content of short-chain fatty acids (SCFAs), such as acetic acid, propionic acid, and butyric acid, which are intricately linked to the health and composition of the intestinal flora. Xue et al. [13] extracted three kinds of soluble dietary fiber from mushroom by-products to explore the effects of different samples on intestinal flora. *Bacteroides* and *Lachnospira* are very suitable for degrading dietary fiber. Mao et al. [14] found that pectin components in dietary fiber significantly increased the abundance of beneficial bacteria (*Bifidobacterium*, *Lactobacillus*) and SCFA-producing bacteria (*Ruminococcus*, *Butyricococcus*), while reducing the relative abundance of *Proteus* and *Actinobacterium.* Fischer et al. [15] found that the relative abundance of *Lachnospiraceae*, *Ruminococceae*, and *Desulfovibrionaceae* has declined in mice raised without dietary fiber. *Alistipes* became the dominant species in mice fed with dietary fiber.

Citrus peel dietary fiber has been found to have antioxidant capacity in previous in vitro experiments, where soluble dietary fiber has a smaller molecular weight and higher antioxidant capacity. Through in vitro intestinal simulation experiments, it was found that SDF could significantly increase the relative abundance of Bacteroidota, reduce the abundance of Firmicutes, and increase the proportion of beneficial bacteria such as *Bacteroides* and *Bifidobacterium*, thereby regulating intestinal microecology. In addition, soluble dietary fiber can significantly increase the content of short-chain fatty acids [16]. Many studies have shown differences in the composition of intestinal flora between young and old people. Changes in intestinal flora are associated with common diseases of old age, such as obesity [17], Alzheimer’s disease, and diabetes [18]. According to the phenomenon that the microbiota changes with age, it is a new potential strategy to intervene in the aging process by altering the intestinal flora [19], because intestinal microecology plays an important role in maintaining intestinal normal function and health, and has become a research hotspot in recent years.

In this study, dietary fiber was extracted from *Citrus unshiu* peel by ultrasound-assisted alkaline pretreatment. A D-galactose-induced oxidative stress model was used to study the effects of dietary fiber on antioxidant and intestinal microecological regulation in mice. The antioxidant capacity of dietary fiber in vivo was analyzed by measuring the activity of antioxidant enzymes and the content of malondialdehyde (SOD, GSH-Px, MDA). The effects of dietary fiber on intestinal microecology in oxidative stress mice were analyzed by measuring intestinal flora and contents of short-chain fatty acids.

## 2. Materials and Methods

### 2.1. Experimental Materials

The mice were housed in an environment with a constant temperature and a 12 h light–dark cycle, with free access to food and water. After a week of acclimatization, the mice were allocated into 7 groups, each containing four mice: a Normal control group (CG), a D-galactose modeling group (MG), a VC treatment group (VC), a PSDF low-dose treatment group (LS), a PSDF high-dose treatment group (HS), a PIDF low-dose treatment group (LI), and a PIDF high-dose treatment group (HI). All groups except the CG received a subcutaneous injection of a D-galactose solution (200 mg/kg/day) for 42 days. From the third week, different treatments were administered orally for 28 days: the CG and MG groups received normal saline, the VC group was given a 100 mg/kg/day Vitamin C solution, the LS and HS groups received PSDF solutions at 100 mg/kg/day and 200 mg/kg/day, respectively, and the LI and HI groups were administered PIDF solutions at the same doses [20]. Mice were weighed and their behavior observed weekly. All animal experiments were performed in compliance with the ethical standards set by the Animal Experiments Ethical Inspection of Zhejiang Center of Laboratory Animals (Permit No. 2023R0007).

### 2.2. Animal Experiment

Male mice, aged 8 weeks, were acquired from Zhejiang Center of Laboratory Animals (Hangzhou, China). D-galactose, ascorbic acid, and short-chain fatty acid standards were sourced from Sigma-Aldrich (St. Louis, MO, USA). GSH-Px, SOD, and MDA detection kits were obtained from Jiancheng Bioengineering Institute of Nanjing (Nanjing, China).

Citrus reticulata was harvested in Xiangshan, Ningbo in October, and the fresh *Citrus unshiu* peel was dried at 50 °C for 48 h, ground into a fine powder using a small mill, and sifted through a 40-mesh sieve. The resulting powder was sealed in an airtight bag and stored in a dry place. This powder was then extracted using ultrasonic assistance with specific parameters: power of 195 W, a liquid-to-solid ratio of 24 mL/g, an NaOH concentration of 0.90%, and an extraction time of 15 min. The supernatant was treated with ethanol to precipitate soluble dietary fiber, while washing the residue yielded insoluble dietary fiber crude products. Soluble dietary fiber was further purified using a macroporous resin method under the conditions of 2 mg/mL sample concentration, 2 bed volumes (BV) sample volume, 70% eluent concentration, and 4 BV eluent volume, achieving a yield of purified soluble dietary fiber (PSDF) of 1.78%. Insoluble dietary fiber underwent purification through enzymatic degradation using amylase (10,000 U/mL), protease (300 U/mL), and saccharifying enzyme (2000 U/mL), resulting in a yield of purified PIDF of 28.64%.

### 2.3. Determination of Antioxidant Capacity

On the final day, mice were fasted for 12 h before sample collection. Blood was drawn from around the eyes and centrifuged at 4 °C at 6000 rpm for 10 min to collect the supernatant. Subsequently, mice were quickly dissected to extract the liver. A 1 g liver sample was homogenized in 9 mL of cold phosphate-buffered saline (PBS), then centrifuged at 4 °C at 4000 rpm for 15 min, and the supernatant was collected for analysis. The levels of malondialdehyde (MDA), glutathione peroxidase activity (GSH-Px), and peroxide dismutase (SOD) in serum and liver homogenate were determined using a commercial colorimetric kit (Nanjing Jiancheng Bioengineering Institute, Nanjing, China), following the manufacturer’s instructions with a microplate reader.

### 2.4. Compositions of Intestinal Flora

Genomic DNA was extracted from fecal samples using kits from Zymo Research Corp (Irvine, CA, USA). The V3-V4 hypervariable regions of the bacterial 16S rRNA gene were amplified and sequenced on the Illumina NovaSeq platform by Beijing Nuohezhiyuan Bioinformatics Co., Ltd. (Beijing, China). The DADA2 plugin in QIIME2 software was used to filter, denoise, merge, and remove chimeric sequences from all raw sample sequences to define operational taxonomic units (OTUs). Species composition, OTU variance between cases, alpha diversity, and beta diversity were assessed for each sample, based on the OTUs’ absolute abundance and annotations.

### 2.5. Concentration of Short-Chain Fatty Acids

The concentration of short-chain fatty acids (SCFAs) in the colons of mice from each intervention group was measured using gas chromatography-mass spectrometry (GC-Trace 1310, MS-ISQ 7000; Thermo Fisher Scientific, Waltham, MA, USA). Mouse fecal samples weighing 50 mg were homogenized in 1 mL of 6% phosphoric acid solution, with 4-methylvaleric acid serving as an internal standard. The column temperature was programmed to start at 90 °C, increase to 120 °C at a rate of 10 °C/min, then to 150 °C at 5 °C/min, and finally to 250 °C at 25 °C/min, where it was held for 2 min. The ion source and MS transfer line temperature were 300 °C and 250 °C, respectively, with an injection volume of 1 μL. Helium was used as the carrier gas with a follow rate of 1 mL/min and a split ratio of 10:1.

### 2.6. Statistical Analysis

All experiments were repeated three times, and the data were presented as the mean ± standard deviation (SD). Data were analyzed statistically using GraphPad Prism 9 (GraphPad Software Inc., San Diego, CA, USA). The significance of differences between mean values was determined through Analysis of Variance (ANOVA), with a significance threshold established at *p* < 0.05.

## 3. Results

### 3.1. Weight Change

Table 1 presents the weight data of mice before and after the experiment. Initially, the mice’s weight ranged from 37 ± 2 g, increasing to 50 ± 3 g post-experiment. Given the variance in starting weights, the analysis focused on weight gain. When comparing each group’s weight gain against the MG, the LS and LI groups exhibited lesser increments, showing a significant difference (*p* < 0.05). This suggests that low doses of PSDF and PIDF contribute to weight reduction during the oxidative stress process in mice.

### 3.2. Antioxidant Capacity of Dietary Fiber In Vivo

During the natural aging process of the human body, the activity of antioxidant enzymes such as superoxide dismutase (SOD) and glutathione peroxidase (GSH-Px) gradually declines, a change that is irreversible. Consequently, SOD and GSH-Px serve as biomarkers for aging and the body’s antioxidant capacity [21]. Additionally, an increase in malondialdehyde (MDA), a secondary product, signifies oxidative stress and lipid peroxidation caused by free radicals.

Figure 1A–C displays the differences in antioxidant enzyme activity and MDA content across various mouse groups, highlighting significant variations when compared to the MG. The MG exhibited the highest MDA concentration at 11.78 nmol/mL. In contrast, the CG (Control Group) presented a substantially lower MDA level at 8.82 nmol/mL (*p* < 0.001), indicating that the D-galactose injection compromised mouse health and accelerated oxidative stress. The VC (Vitamin C Group) recorded an MDA level of 9.81 nmol/mL, positioning it between the MG and CG. The LS (Low-dose PSDF Group) and HS (High-dose PSDF Group) demonstrated MDA levels of 9.90 nmol/mL and 9.62 nmol/mL, respectively, both notably lower than the MG (*p* < 0.05). This suggests a dose-dependent improvement in MDA levels attributed to PSDF, with no significant difference between the LI and HI group.

Antioxidant enzyme activities (SOD and GSH-Px) were reduced in the MG to 430.10 U/mL and 192.42 U/mL, respectively, whereas the CG showed higher activities at 520.86 U/mL and 265.57 U/mL, confirming the oxidative stress induced by D-galactose. The LS group exhibited a significant increase in SOD activity to 496.23 U/mL (*p* < 0.05), with the VC group also showing enhanced enzyme activities.

The antioxidant enzyme activity and MDA content analysis confirm the CG’s superiority over the MG, establishing an oxidative stress mouse model. VC was effective even at low doses, and various doses of PSDF demonstrated differing antioxidant capabilities, notably with low doses effectively reducing MDA levels. These findings underscore the potential of dietary fiber to mitigate oxidative damage from D-galactose in mouse serum.

#### Analysis of Antioxidant Enzyme Activity and MDA Content in Liver

Figure 1D–F illustrates the variations in antioxidant enzyme activity and MDA content in the liver of mice across different groups, with a comparative analysis to the MG.

The MDA levels were highest in the MG at 1.94 nmol/mgprot, whereas the CG showed a lower concentration at 1.14 nmol/mgprot, confirming liver damage due to D-galactose. The VC group, LS group, and HS group exhibited MDA levels of 1.47, 1.48, and 1.37 nmol/mgprot, respectively, indicating a consistent regulation trend across liver and serum. No significant differences were observed among other groups.

SOD (Superoxide Dismutase) activities for the MG, CG, VC, LS, and HS groups were 720.18, 910.17, 866.24, 931.79, and 884.34 U/mgprot, respectively, highlighting the LS group’s superior enhancement of SOD activity in oxidative stress mice. GSH-Px activities were 523.48, 611.41, 600.06, 593.60, and 630.29 U/mgprot for the same groups, with the HS group showing the most significant improvement in GSH-Px activity. No notable differences were found in antioxidant enzyme activities between the LI group and HI group. This study indicates that PSDF doses variably boost antioxidant enzyme activities, with low-dose PSDF favoring SOD and high-dose PSDF benefiting GSH-Px activities.

### 3.3. Effects of Dietary Fiber on Intestinal Flora

#### 3.3.1. Dietary Fiber Regulates the Intestinal Flora Diversity and Structure in D-Galactose-Induced Oxidative Stress Mice

To demonstrate the impact of dietary fiber supplementation on the diversity and structure of intestinal flora in oxidative stress mice induced by D-galactose, we conducted 16S rRNA sequencing on fecal samples collected on the study’s final day. Analysis via a Venn diagram revealed 317 Operational Taxonomic Units (OTUs) shared across all seven experimental groups, with unique OTUs numbering 69, 165, 212, 132, 142, 60, and 230 for the MG, CG, VC, LS, HS, LI, and HI groups, respectively (Figure 2). Notably, mice receiving high doses of PIDF exhibited a greater OTU count than other groups (Figure 2A). Alpha diversity, serving as a fundamental metric for intestinal flora diversity across the seven mouse groups, indicated no significant differences in the Chao1 and Shannon indexes among the MG, LS, HS, LI, and HI groups (*p* > 0.05) (Figure 2B,C). The MG, when compared to the control (CG), showed a marked reduction in microbial richness and diversity, implying that D-galactose administration could compromise intestinal flora health. Conversely, all intervention groups, with the exception of the LI group, managed to enhance microbial richness to varying degrees.

Based on the operational taxonomic units, Principal Coordinates Analysis (PCoA) using Bray–Curtis distance assessed the variations among the samples from each treatment group (Figure 2D). Notably, within the same population, there was considerable variability in species abundance. This study highlighted significant variances in the colonic microbiome across different populations. According to the figure, the PC1 and PC2 accounted for 67.03% and 7.87% of the variation, respectively. A comparison between each group and the MG revealed greater distance in the centroids of the LS and VC groups. In contrast, the centroids of the LI and HS groups were closely positioned, suggesting minimal variation between them. These findings suggest that low doses of PSDF and VC significantly disrupt the intestinal flora in oxidative stress mice, leading to pronounced differences among the samples.

#### 3.3.2. The Regulatory Effect of Dietary Fiber on Intestinal Flora Composition in D-Galactose-Induced Oxidative Stress Mice

The utilization of bar chart visualizations further elucidated the changes in intestinal flora after dietary fiber treatment (Figure 3A,B). Significant disparities were evident in intestinal flora across the seven distinct mouse populations. Analysis confirmed that Bacteroidota, Firmicutes, Actinobacteriota, and Proteobacteria were the predominant microbial groups, with Bacteroidota and Firmicutes together representing over 80% of the total microbial composition. The abundance of Bacteroidota in the MG, CG, VC, LS, HS, LI and HI groups was 69.85%, 60.29%, 38.09%, 49.52%, 62.09%, 80.52%, and 51.26%, respectively. The abundance of Firmicutes was 25.14%, 29.52%, 42.42%, 44.48%, 29.54%, 15.90%, and 38.51%, respectively. Relative to the control group, an increase in Bacteroidota and a decrease in Firmicutes, Proteobacteria, and Actinobacteriota were noted in the MG. Compared with the MG, the level of Bacteroidota in the LI group was higher, and that in the VC group, LS group, HS group, and HI group was lower. Conversely, Firmicutes levels were generally higher in the groups other than LI. Proteobacteria levels increased in the VC, HS, and HI groups but decreased in the LS and LI groups. At the genus level, significant differences were observed; for example, the VC group showed a substantially lower abundance of *Muribaculaceae* (24.43%) compared to the MG (43.08%), as highlighted in Figure 3B. Additionally, *Pseudomonas* and *Lactobacillus* were more prevalent in the VC group (3.44% and 9.27%, respectively) than in the MG (0.40% and 1.38%). *Lachnospiraceae_NK4A136_group* was much more abundant in the LS group (12.49%) than in the MG (3.37%). *Alloprevotella* was found in greater abundance in the LI group (12.72%) compared to the MG (8.84%). *Lactobacillus* was more prevalent in the LS and HI groups (2.28% and 7.76%, respectively) than in the MG (1.38%). *Pseudomonas* was far more abundant in the LS, HS, and HI groups (0.59%, 1.42%, and 2.10%, respectively) than it was in the MG (0.40%). The HI and HS groups had significantly larger abundances of *Akkermansia* (1.90% and 1.88%, respectively) than the MG (0.08%).

Linear Discriminant Analysis Effect Size (LEfSe) and Linear Discriminant Analysis (LDA) were utilized to evaluate the impact of the abundance of individual species on differential outcomes, aiming to identify significant groups or species that contribute to the distinctiveness of samples (Figure 3C,D). It showed *s_Bacteroides_dorei* and *g_Staphylococcus* were identified as the dominant microbiota in the MG. The LS group appeared to be dominated by *p_Firmicutes*, *g_Lachnospiraceae NK4A136_group*, and *g_Eubacterium_xylanophilum_group*, whereas the HS group showed *f_Oscillospiraceae*, *c_Verrucomicrobiae*, *p_Verrucomicrobiota*, *o_Verrucomicrobiales*, *g_Akkermansia*, *f_Akkermansiaceae*, and *g_Prevotellaceae_UCG_001* as the dominant microbiota. *c_Bacteroidia*, *o_Bacteroidales*, *p_Bacteroidota*, *f_Prevotellaceae*, and *g_Alloprevotella* were identified as the dominant microbiota in LI group. And the HI group appeared to be dominated by *c_Bacilli*, and *o_Pseudomonadales*.

### 3.4. SCFAs Analysis

Gas Chromatography-Mass Spectrometry (GC-MS) detected seven types of short-chain fatty acids (SCFAs) in the colons of mice: propionic, butyric, caproic, valeric, acetic, isobutyric, and isovaleric acids (Figure 4). The MG served as the control for comparison with other groups to analyze intergroup differences. SCFA levels in the CG group were generally higher than in the MG, with caproic acid showing a significant difference. Excluding the CG, the LI group exhibited a caproic acid concentration of 148.01 μg/mL, significantly surpassing the MG, potentially due to an increased presence of *Bacteroidetes* and *Prevotella*.

The LS group demonstrated superior SCFA regulation, with notably higher levels of acetic acid (223.44 μg/mL), butyric acid (37.23 μg/mL), and valeric acid (5.35 μg/mL) compared to the MG. This group also matched the CG in intestinal flora diversity, suggesting that supplementing oxidative stress mice with low-dose polysaccharide dietary fiber (PSDF) beneficially impacts intestinal diversity restoration. Furthermore, the HS group’s butyric acid concentration was 35.10 μg/mL, significantly exceeding the MG’s levels, indicating the varied impact of PSDF doses on SCFA levels.

### 3.5. Correlation Analysis of Intestinal Flora, SCFAs, and Antioxidant Capacity of Dietary Fiber

Spearman correlation analysis was utilized to examine the relationships among intestinal flora changes, metabolites, and antioxidant markers, focusing on the relative abundance of the top 20 genera, four principal phyla, short-chain fatty acids (SCFAs), and serum and liver levels of malondialdehyde (MDA), superoxide dismutase (SOD), and glutathione peroxidase (GSH-Px) as depicted in Figure 5. Our findings, presented in Figure 5A, showed that there was a significant positive correlation between Firmicutes and isovaleric acid; *Helicobacter* was significant positive correlation between acetic acid and propionic acid. Conversely, *Muribaculaceae* was negatively correlated with acetic acid. Additionally, Figure 5B indicated that *Sporosarcina* was positive correlated with MDA in serum. However, Figure 5C demonstrated that liver SOD levels were significantly positively correlated with butyric acid, isovaleric acid, and isobutyric acid. Studies have shown that dietary fiber supplementation can increase the content of short-chain fatty acids in the intestinal tract of colitis rats, and also improve the activity of antioxidants such as SOD [22].

## 4. Discussion

In oxidative stress mice, soluble dietary fiber (PSDF) was found to significantly enhance the activity of antioxidant enzymes and reduce malondialdehyde (MDA) concentrations, while insoluble dietary fiber (PIDF) exhibited no noticeable regulatory effect. PSDF’s efficacy varied, being less potent in serum but more so in the liver, and its regulatory capacity differed across dosages, indicating concentration-dependent anti-oxidative stress benefits. The role of intestinal tract probiotics in human health is pivotal, influencing aspects such as enhancing the epithelial barrier, modulating mucosal inflammation, and preserving intestinal flora balance. Figure 3A demonstrates that Bacteroidota and Firmicutes constitute the primary bacteriophyt in mice, representing over 80% of total intestinal flora. Firmicutes and Bacteroidota are the two most abundant bacteria in the intestine. The ratio of the two can indicate the extent of intestinal disorder. For example, excessive alcohol consumption can lead to an increase in Firmicutes and a decrease in Bacteroidota [23]. Moreover, colitis mice exhibited diminished blood antioxidant activity correlating with an increased Firmicutes and decreased Bacteroidota levels. Melatonin supplementation has been shown to enhance the intestinal flora through the activation of antioxidant enzymes [24]. Excessive D-galactose accumulation and subsequent oxidative damage to the gastrointestinal mucosa can disrupt intestinal microecology [25]. Research indicates that an elevated Bacteroidota to Firmicutes ratio (B/F index) in mice’s intestines is a consequence of excess D-galactose [26], a finding aligned with this study’s results.

The B/F index in the control group (CG) was lower than in the model group (MG), with the exception of the LI group, where the B/F ratio in all other treatment groups also fell below that of the MG. These findings suggest that Vitamin C (VC) and dietary fiber can counteract the alterations in the predominant intestinal flora observed in oxidative stress mice, thereby aiding in the restoration of intestinal flora balance. Moreover, the B/F ratio appears to be linked to obesity, with a higher B/F ratio correlating with a reduced obesity risk. Analysis of intestinal flora composition revealed that the B/F ratio in the LI group was significantly elevated compared to other groups. Concurrently, data indicated that the body weight of mice in the LI group was notably lower than those in the MG. This suggests that the ratio of Bacteroidota to Firmicutes might influence the body weight of mice.

*Muribaculaceae* is a type of intestinal probiotic mucomyces, a predominant genus within the intestinal tract of mice, which is rich in enzyme systems for carbohydrate breakdown, can metabolize short-chain fatty acids (SCFAs), and promote the growth and development of intestinal epithelial cells (ICEs). Its abundance, however, diminishes in mouse models presenting conditions such as immunosuppression, intestinal damage, obesity, hyperglycemia, and metabolic syndrome [27]. *Lactobacillus*, conversely, is recognized for its beneficial impact on enhancing antioxidant capabilities, with a positive correlation observed between its abundance and antioxidant levels. Notably, *Lactobacillus* exhibits an elevated ability to scavenge DPPH radicals, demonstrating resistance to oxidative stress induced by D-galactose [19]. These observations suggest that while the introduction of Vitamin C (VC) may disturb the equilibrium of intestinal flora, the augmented presence of *Lactobacillus* could be associated with improved antioxidant functions.

Members of the *Lachnospiraceae_NK4A136_group* are also capable of metabolizing mucin monosaccharides. In a study, diabetic mice, induced by a high-fat diet and streptozotocin, received intragastric administration of purified Cordyceps militaris polysaccharide. This treatment led to an increased abundance of the *Lachnospiraceae_NK4A136_group*, which in turn inhibited the toll-like receptor 4 (TLR4)/nuclear factor κB (NF-κB)-signaling pathway in the colon [28]. The sulfated polysaccharides in asparagus may accelerate the conversion of cholesterol to bile acids through gene expression and regulation of intestinal flora. The relative abundance of the *Lachnospiracae_NK4A136_group* was positively correlated with some hydrophilic bile acid levels [29]. Therefore, the increase in the *Lachnospiraceae_NK4A136_group* may reduce the incidence of obesity. At the same time, glucose tolerance and lipid levels are also affected. Combined with the above results, the weight loss of the LS group may be related to the increased abundance of the *Lachnospiraceae_NK4A136_group*. In summary, Vitamin C (VC) administration to the intestinal flora does not enhance dominant bacterial populations, and may instead lead to an increased abundance of pathogenic bacteria, suggesting its potential impact on intestinal immune function. Low doses of PSDF have been shown to increase the abundance of the *Lachnospiraceae_NK4A136_group* and positively regulate metabolism in oxidative stress mice.

Furthermore, *Alloprevotella* has been identified as negatively correlated with antioxidant enzyme activities (SOD and GSH-Px) and GSH levels, highlighting its role in oxidative stress [30]. Distinctly, *Proteobacteria* levels in the CG differ from those in other groups, with this phylum being closely associated with various diseases and containing numerous human pathogens that may contribute to intestinal inflammation and metabolic disorders [31]. However, dietary fiber supplementation appears to preserve intestinal immunity in mice, without promoting *Proteobacteria* growth, thereby potentially restraining pathogenic bacteria proliferation.

Short-chain fatty acids (SCFAs), primarily produced by intestinal flora from undigested carbohydrates and some proteins, include acetic, propionic, and butyric acids, constituting about 90–95% of total intestinal SCFAs. This study observed a notable increase in caproic acid levels following low-dose PIDF supplementation, while low-dose PSDF supplementation significantly boosted levels of acetic, butyric, and valeric acids. High-dose PSDF intervention notably enhanced butyric acid concentrations, with PSDF treatment maintaining intestinal flora diversity akin to that of the normal group, indicating a beneficial effect on restoring intestinal flora diversity. Studies have shown that butyric acid can activate peroxisome proliferator activator receptor-γ in colon cells, inhibit NOS_2_ expression, and reduce nitrate levels [32]. Many polysaccharides play a regulatory role in SCFAs. Cyclocarya paliurus polysaccharide can significantly increase the contents of acetic acid, propionic acid, butyric acid and valerate in the feces of healthy mice [33]. Mulberry leaf-derived polysaccharide can reduce the oxidative damage of the liver and restore normal levels of acetic acid, propionic acid, and butyric acid [34]. Tangerine peel powder increased the ratio of Bacteroides to Firmicutes in the intestines of mice, and slightly increased concentrations of acetic acid, valerate, and butyric acid. Since the production of short-chain fatty acids is closely related to the composition of intestinal flora [35], both the HS and LS groups showed strong antioxidant capacity and intestinal regulation ability. Subsequently, results based on correlation analysis showed a relationship between intestinal flora, SCFAs, and the antioxidant index at genus and phylum levels. The results showed that Firmicutes were positively correlated with isovaleric acid, *Helicobacter* was positively correlated with acetic acid and propionic acid, *Muribaculaceae* was negatively correlated with acetic acid, and the *Lachnospiraceae_NK4A136_group* was negatively correlated with caproic acid. In addition, the liver SOD level was positively correlated with butyric acid, isovaleric acid, and isobutyric acid. Li et al. [36] found that the gastric administration of PFP reduced the ratio of Firmicutes/Bacteroidota (F/B), which may be related to the high polysaccharide utilization ability of *Bacteroidota*, and increased the SOD index in the liver of mice. Therefore, we speculate that there may be a certain correlation between intestinal flora and antioxidant effect, which needs to be further explored.

In summary, soluble dietary fiber effectively enhances SOD and GSH-Px activities and significantly reduces MDA levels, unlike insoluble dietary fiber, which shows limited antioxidant capacity enhancement. Polysaccharides combat oxidative damage by increasing the activities and mRNA expression of antioxidant enzymes [37]. Oxidative stress diminishes the mRNA expression of SOD and GSH-Px, implying the regulation of antioxidant enzymes in oxidative stress tissues through translational and post-translational modifications [38]. Therefore, the impact of soluble dietary fiber on these enzymes might also involve their translation processes. The exact antioxidant mechanisms remain to be further investigated.

## 5. Conclusions

This study analyzed the impact of dietary fiber from *Citrus unshiu* peel on antioxidant enzyme activities and MDA levels in an oxidative stress mouse model, examining the link between oxidative stress and intestinal ecology through microflora and SCFAs analyses. It was discovered that PSDF significantly enhanced antioxidant enzyme activities while reducing MDA levels in oxidative stress mice, with varying antioxidant effects observed across different PSDF dosages. In contrast, PIDF exhibited no notable regulatory effects. 16S rRNA amplification sequencing revealed that low-dose PSDF improved intestinal flora diversity and increased the abundance of the *Firmicutes* and *Lachnospiraceae_NK4A136_group* in oxidative stress mice. Both low and high doses of PSDF significantly elevated the levels of acetic acid, butyric acid, and valerate. Meanwhile, a low dose of PIDF was found to boost the abundance of *Bacteroidota* and *Alloprevotella*, enhancing caproic acid production. Compared to dietary fiber treatments, VC increased intestinal flora diversity but did not significantly influence SCFAs levels. Thus, this research bridges the knowledge gap regarding the antioxidant and intestinal regulatory effects of PIDF and PSDF in *Citrus unshiu* peel in oxidative stress mice, suggesting their potential as functional components in the food industry.

## Figures and Tables

**Figure 1 foods-13-01539-f001:**
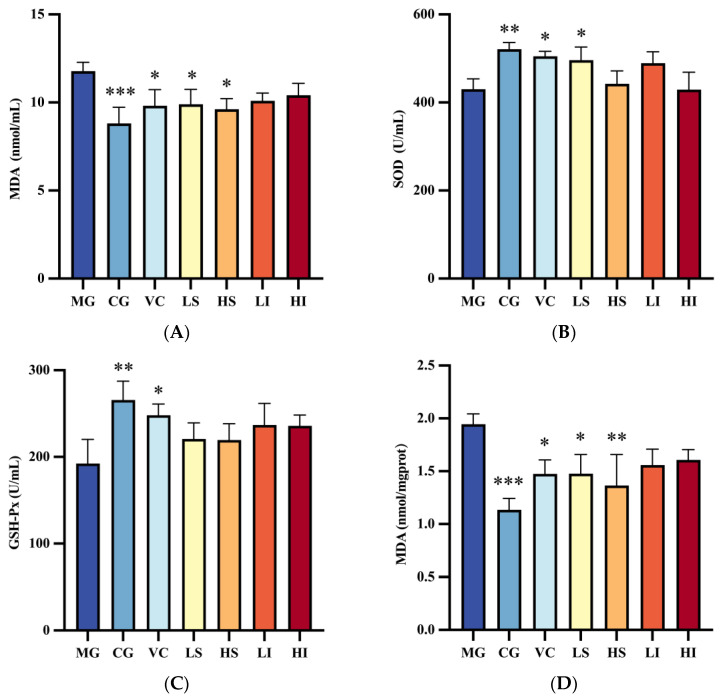
Effects of dietary fiber PSDF and PIDF on the production of malondialdehyde (MDA), superoxide dismutase (SOD), and glutathione peroxidase (GSH-Px) in serum and liver of D-galactose-induced oxidative stress mice. (**A**) MDA in serum, (**B**) SOD in serum, (**C**) GSH-Px in serum, (**D**) MDA in liver, (**E**) SOD in liver, (**F**) GSH-Px in liver. Data are presented as mean ± SD. According to the ANOVA analysis, the differences between each group and the MG were analyzed. * *p* < 0.05, ** *p* < 0.01, and *** *p* < 0.001.

**Figure 2 foods-13-01539-f002:**
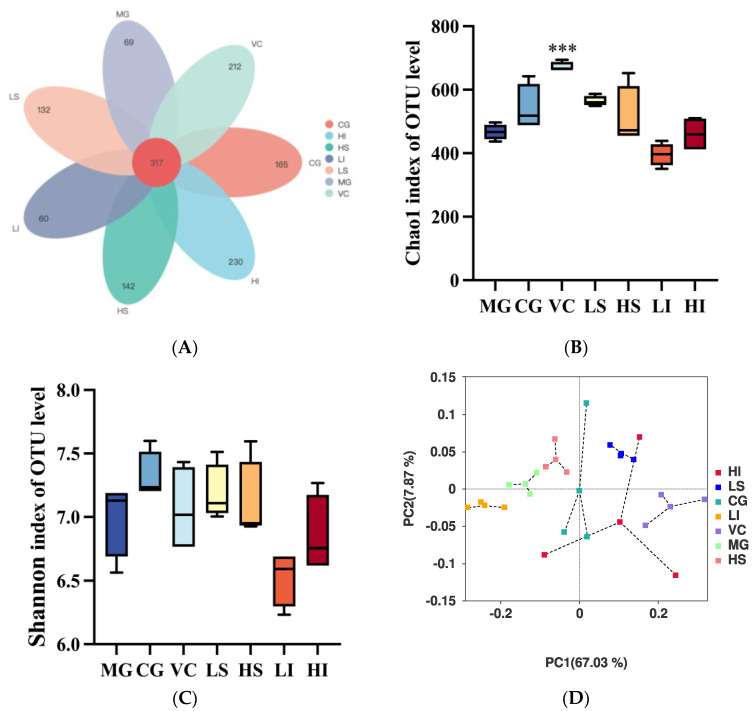
Effects of dietary fiber on intestinal flora diversity and structure in D-galactose-induced oxidative stress mice. (**A**) The Venn diagram shows the intestinal flora of different groups based on operational taxonomic unit (OTU) values, (**B**) Chao1 index of all groups, (**C**) Shannon index of all groups, (**D**) PCoA analysis. Data are presented as mean ± stand deviation (*n* = 4). *** *p* < 0.001.

**Figure 3 foods-13-01539-f003:**
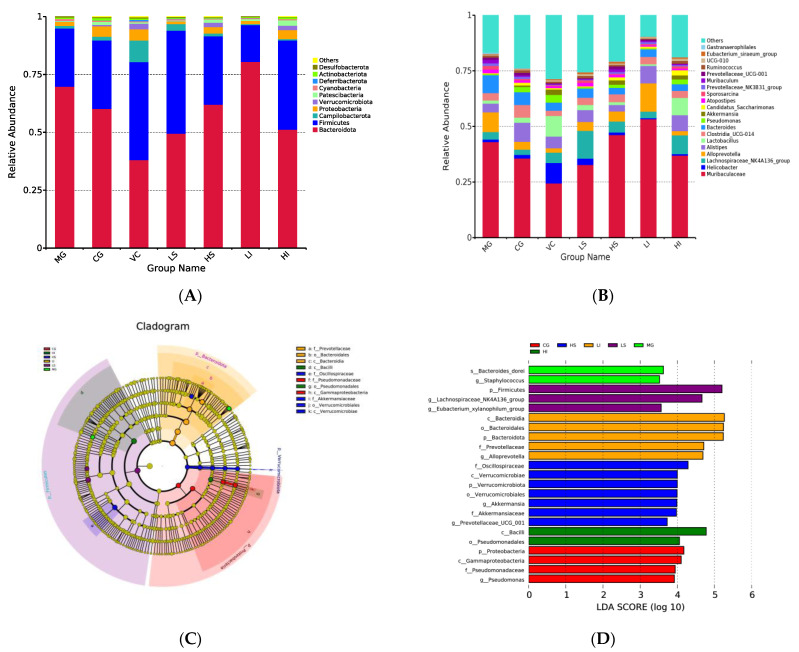
The regulatory effect of dietary fiber on intestinal flora composition in mice. (**A**) Phylum level, (**B**) genus level, (**C**) linear discriminant analysis effect size, (**D**) linear discriminant analysis (LDA; LDA score > 3.5).

**Figure 4 foods-13-01539-f004:**
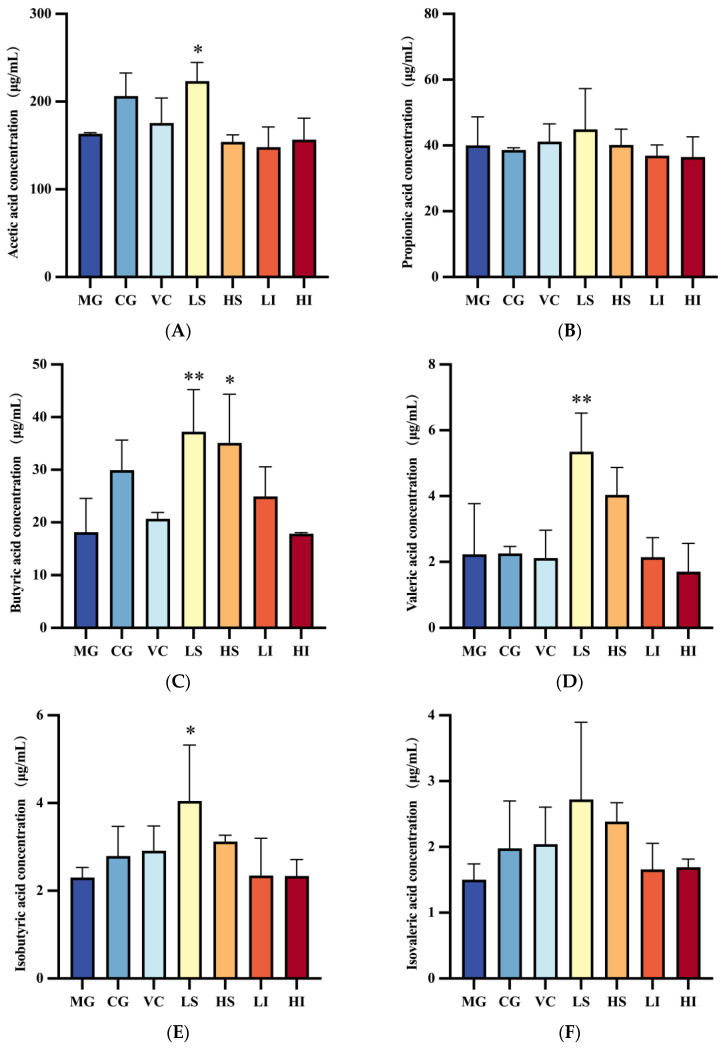
Effects of dietary fiber PSDF and PIDF supplementation on intestinal SCFA production in mice. (**A**) Acetic acid, (**B**) propionic acid, (**C**) butyric acid, (**D**) valeric acid, (**E**) isobutyric acid, (**F**) isovaleric acid. Data are presented as mean ± SD. According to the ANOVA analysis, * *p* < 0.05, ** *p* < 0.01.

**Figure 5 foods-13-01539-f005:**
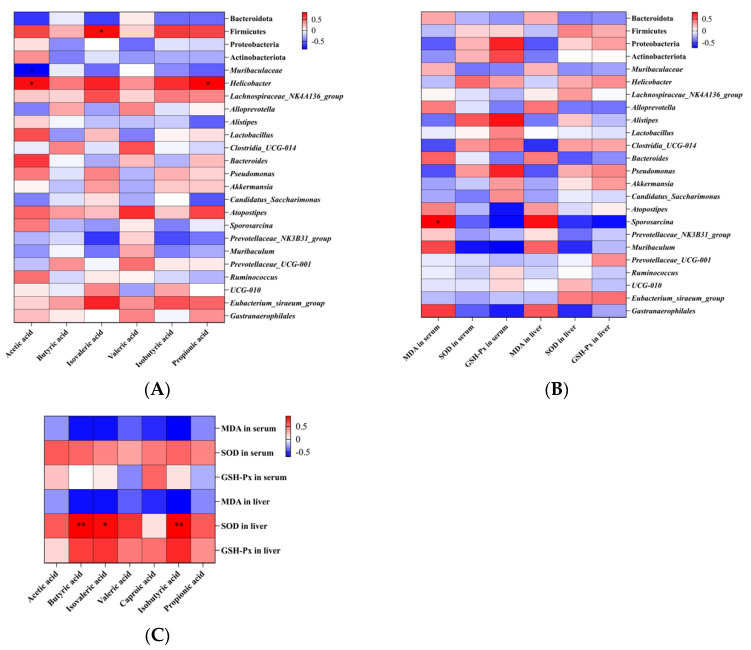
Spearman correlation of intestinal flora (top 20 genus level and top 4 phylum level), SCFAs and antioxidant indexes. (**A**) Correlation between intestinal flora and SCFAs. (**B**) Correlation between antioxidant indexes and intestinal flora. (**C**) Correlation between antioxidant indexes and SCFAs. Red indicates a positive correlation and blue indicates a negative correlation. * *p* < 0.05, ** *p* < 0.01.

**Table 1 foods-13-01539-t001:** Changes of body weight in mice.

Group	Body Weight (g)		Weight Gain (g)
	Beginning	Ending	
MG	36.78 ± 0.79	51.91 ± 1.70	15.11 ± 0.96 *
CG	36.72 ± 1.13	51.23 ± 2.11	14.49 ± 2.23 *
VC	38.35 ± 0.26	52.16 ± 1.09	13.81 ± 0.93 *
LS	36.31 ± 0.68	48.48 ± 1.34	12.17 ± 0.73 *
HS	38.73 ± 0.70	52.07 ± 0.33	13.34 ± 0.39 *
LI	35.8 ± 0.67	47.20 ± 0.90	11.40 ± 0.30 *
HI	36.69 ± 0.56	52.46 ± 2.15	15.77 ± 1.88 *

Note: ANOVA analysis was performed for each group and MG group, and there are significant differences between the data (* *p* < 0.05).

## Data Availability

The data presented in this study are available on request from the corresponding author due to privacy.

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
