# Peer review of "Soluble Dietary Fiber from Citrus unshiu Peel Promotes Antioxidant Activity in Oxidative Stress Mice and Regulates Intestinal Microecology"

_foods, 2024, doi:10.3390/foods13101539_

Round 1

Reviewer 1 Report

Comments and Suggestions for Authors

Dear authors,

The aim of the study of this study is to assess the antioxidant activity of the peel of Citrus unshiu. The whole study is excellent but I am not sure if its subject is suitable for "Foods". From the information that the authors provide in the introduction, the peel of the Citrus unshiu is neither an edible part of the fruit nor it is used in any culinary way. 

The introduction is very informative except for the last lines 103-111, which are obviously an editing accident. In the introduction the final lines 103-111 should be deleted and replaced by the aim of the study.

The experimental design is very robust. Experimental animals are divided in seven groups and apart from the controls, all other groups undergo a special treatment with D-galactose as an oxidizing aging factor. Administration of soluble and insoluble fibers from the peel of Citrus unshiu in high and in low doses followed. The weight gain was measured among the different groups.  Fecal samples, blood samples and liver samples were collected and measurments of biomarkers such as SOD and GSH-Px enzymes as well as the MDA enzyme were measured in the serum and in the liver tissue. The biodiversity of the intestinal flora was also assessed. Short chain fatty acids as a product of the intestinal fermentations were also measured.

The results are well presented with clarity, while graphs and figures make them more comprehensive. The antioxidizing effect of the soluble fibers was impressive. The biodiversity of the intestinal flora was seriously affected as the ratio B/F (bacteroida/firmicutes) shows as well as the SCFAs levels.

The discussion interpretes thye rsults by comparing them to findings of other researchers.

The conclusiuons are directly derived by the findings.

Author Response

Thank you for your suggestion. Citrus peel in China is a traditional medicine and food cognate food, usually we call it chenpi. The section on editing errors has been removed.

Reviewer 2 Report

Comments and Suggestions for Authors

I reviewed the article entitled "Soluble dietary fiber from Citrus unshiu peel promotes antioxidant activity in oxidative stress mice and regulates intestinal microecology", written by Mengqi Fu, Xin Gao, Zuorui Xie, Chenlan Xia, Qing Gu, and Ping Li.

The purpose of this manuscript is to determine whether dietary fiber from Citrus unshiu possesses beneficial antioxidant and regulatory activities in the gut microbiota in vivo.

The scientific collection is very interesting. However, some problems indicated below should be solved. This version of the manuscript is not complete. Below, I present my objections/suggestions in detail.

General remarks:

·       What is the meaning of "#" in authors' names? An equal contribution?

Abstract:

·       I recommend better highlighting the purpose of the work and the conclusions.

Introduction:

·       Lines 33-35: In today's context, the proposal to valorize a product thought to be a waste product by exploiting it as a source of antioxidant substances is fascinating. The same suggestion has already been formulated by other authors, with successful results. I suggest the authors add a comment on the following manuscript: (doi: 10.3389/fphys.2023.1225552).

·       Lines 103-111: Remove this part.

Materials and methods:

·       Line 122: Why was vitamin C considered? As the antioxidant gold standard?

·       Line 123: Why was this fiber concentration selected?

·       Line 133: It would be appropriate to indicate the place and time of the Citrus harvest.

Results:

·       Lines 238-244: This part should be moved to the Discussion section.

·       Line 343, Table 1: In the other results, significance was denoted by the symbol "*". I suggest that the table should also be unified.

·       Line 356: “*p<0.05” should be modified in “***p<0.001”.

·     Line 357, Figure 3: Figures 3a and 3b should be magnified. In Figures 3c and 3d, the resolution should be implemented.

Discussion:

·       Line 378: Delete this sentence; it is pleonastic.

·       Line 404-406: Please check this sentence.

·     Lines 425-427: Similar results on the action of vitamin C can be found in other recent studies, such as the following: (doi: 10.1002/jbt.22986). I suggest that the authors add some information about this.

Conclusion:

·       The conclusion is well-written and exhaustive in all its parts, I have no suggestions.

References

·       The references of this work are few. I suggest implementing this section.

Comments on the Quality of English Language

A little spell-checking is necessary to ensure that an international audience can clearly understand the text. In general, I suggest revising the style of the manuscript according to the guidelines of the journal.
